# Biofunctionalization of Textile Materials. 2. Antimicrobial Modification of Poly(lactide) (PLA) Nonwoven Fabricsby Fosfomycin

**DOI:** 10.3390/polym12040768

**Published:** 2020-04-01

**Authors:** Marcin H. Kudzin, Zdzisława Mrozińska

**Affiliations:** The Lukasiewicz Research Network -Textile Research Institute, Brzezinska 5/15, 92-103 Lodz, Poland; zmrozinska@iw.lodz.pl

**Keywords:** poly(lactide) nonwoven fabric, fosfomycin, phosphomycin disodium salt, phosphonomycin, antibacterial activity, polymer functionalization, polyamide

## Abstract

This research is focused on obtaining antimicrobial hybrid materials consisting of poly(lactide) nonwoven fabrics and using phosphoro-organic compound—fosfomycin—as a coating and modifying agent. Polylactide (PLA) presents biodegradable polymer with multifunctional application, widely engaged in medical related areas. Fosfomycin as functionalized phosphonates presents antibiotic properties expressed by broad spectrum of antimicrobial properties. The analysis of these biofunctionalized nonwoven fabrics processed by the melt-blown technique, included: scanning electron microscopy (SEM), UV/VIS transmittance, FTIR spectrometry, air permeability. The functionalized nonwovens were tested on microbial activity tests against colonies of gram-positive (*Staphylococcus aureus*) and gram-negative (*Escherichia coli*) bacteria.

## 1. Introduction

Polylactide (PLA) presents polymers of multifunctional application, widely used in medical related areas [1]. PLA hybrids with antibacterial additivities (bactericide agents) provide antiseptic properties, and therefore are applied in a variety of medical applications, namely, such as bioactive fibers for drug delivery [2,3,4,5,6,7,8,9,10,11,12,13,14,15,16,17,18], for tissue engineering applications [19,20,21,22], forwound healing materials [23,24,25,26] and membranes for efficient treatment of burns [27,28].

As one group of such potential additives/components, they can serve as functionalized phosphonates. Functionalized phosphonate compounds, especially amino-phosphonates, are known for their significant biological activity [29]. The list of representative biologically active phosphonates is presented in Table 1.

Fosfomycin (cis-1,2-epoxypropylphosphonic acid) exhibits broad spectrum activity against numerous bacterial, both gram-positive and gram-negative pathogens, including resistant and multi-resistant strains (Table 1).Fosfomycin acts as a bacterial cell wall inhibitor in the bacteria growth phase, interfering with the first committed step in peptidoglycan biosynthesis. Specifically, Fosfomycin irreversibly inhibits the UDP-N-acetylglucosamine enolpyruvyl transferase (MurA) enzyme by covalent modification of MurA (Figure 1), responsible for catalyzing the formation of N-acetylmuramic acid (precursor of peptidoglycan) [54].

Due to the fact that Gram-positive and Gram-negative bacteria require the formation of N-acetylmuramic acid for the synthesis of peptidoglycan, fosfomycin’s antibacterial spectrum is very broad, and there is no possibility of crossed resistances with this compound. This antibiotic has therefore been employed for treating infections by multidrug-resistant pathogens such as methicillin-resistant *Staphylococcus aureus* (MRSA), methicillin-resistant coagulase-negative staphylococci (MRCNS), vancomycin-resistant enterococci (VRE), penicillin-resistant Streptococcus pneumoniae, extended-spectrum beta-lactamase (ESBL)-producing *Enterobacterales*, carbapenemase-producing *Enterobacterales* (CPE) and multidrug-resistant *Pseudomonas aeruginosa* [51].

Fosfomycin was accepted into clinical practice in the early 1970s [54,56]. Its use, however, has been limited for several years for treating mainly lower uncomplicated urinary tract infections (in the form of fosfomycin trometamol taken orally). Therefore, the application of fosfonomycin as an additive to various PLA medical utilities seems to be highly rational. However, the recent paper of Gulcuet al. revealed that the application of the PDL-fosfomycin hybrid used by authors in implant coating for the prevention of implant-related infections afforded worse results than corresponding hybrids based on PLA-gentamycin [58]. To dissolve these discrepancies, we undertook wide investigations on polymer hybrids, based on polylactide (PLA) nonwoven fabric modified on the surface by fosfomycin (FOSM).

As part of our research program directed on biologically active functionalized phosphonates [59,60,61,62,63], and their grafting on polymer matrix [64,65], we present the preparation, and physico-chemical and biological properties of PLA–FOS polymer hybrid.

## 2. Materials and Methods

### 2.1. Materials

#### 2.1.1. Polymers

Poly(lactic acid) (PLA) granulate was purchased from NatureWorks LLC (Minnetonka, Minnesota, USA), type Ingeo™ Biopolymer 3251D, MFR = 30—40 g/10min (190 °C/2.16 kg), T_mp_ = 160—170 °C in the form of granulates, and was used for the fabrication of nonwoven samples.

#### 2.1.2. Chemical Agents

Fosfomycin[(−)-(1R,2S)-(1,2-Epoxypropyl)phosphonic acid,C_3_H_5_Na_2_O_4_P] from Sigma-Aldrich (St. Louis, MI, USA) was used for the surface modification of polymer nonwovens.

#### 2.1.3. Finishing Agents

Lutexal Thickener HC (BASF, Ludwigshafen, Germany) – polyacrylate, ammonia salt as the thickening agent, ensures the viscosity and appropriate grip of product;

Pluriol 600 (BASF, Ludwigshafen, Germany) - poly(ethylene glycol) of molar mass 600 g/mol as the wetting agent, prevents the formation of agglomerates in coating paste and uniformity in coating dispersion;

Revacryl 247 (Synthomer, Essex, UK)- dispersion of low viscosity styrene-acrylic ester copolymer, combines all ingredients of coating paste with nonwoven fabric.

#### 2.1.4. Bacterial Strains

*Escherchia coli* (ATCC 25922); *Staphylococcus aureus* (ATCC 6538) bacterial strains were purchased from Microbiologics (St. Cloud, MN, USA).

### 2.2. Methods

#### 2.2.1. Nonwoven Fabrics

Poly(lactic acid) nonwovens were fabricated by the melt-blown technique, analogously as polypropylene nonwovens [65], using a one-screw laboratory extruder (Axon, Sweden) with a head with 30 holes of 0.35 mm diameter each, compressed air heater and collecting drum. Processing parameters for fabrication of poly(lactic acid) nonwoven are presented in Table 2.

#### 2.2.2. Dip-Coating Modification of Poly(lactic acid) Nonwoven

Poly(lactic acid) nonwoven fabrics were modificated by the dip-coating method, analogously as polypropylene nonwovens [65]. The coating pastes of homogeneous dispersion and appropriate viscosity (70 dPas) were prepared based on styrene-acrylic resin, thickening agent, wetting agent and water. Component composition of used pastes is listed in Table 3.

Fosfomycin powder was added into the paste in 3 variants of concentrations: 0.005%, 0.01%, 0.1%and then mixed for 10 min.

The poly(lactic acid) (PLA) nonwoven samples were impregnated with the paste, squeezed and dried to the constant weight for 5 h at a temperature of 50 °C. The increase of sample dry mass after modification was 15%.

#### 2.2.3. SEM—Scanning Electron Microscopy

The scanning electron microscope analysis was performed on a TESCAN VEGA 3 SEM microscope (Czech Republic). The SEM microscopic examination of the surface topography was carried out in a high vacuum using the energy of the probe beam 20 ekV. The surface of each preparation was sprayed with a conductive substance (gold), using a Quorum Technologies Ltd. (UK) vacuum dust extractor. Magnification was 100×, 5000× and 20,000×.

#### 2.2.4. ATR-FTIR—Attenuated Total Reflection Fourier Transform Infrared Spectroscopy

The chemical structure of poly(lactic acid) surface of nonwovens were assessed using ATR-FTIR spectroscopy in the range of 400–4000 cm^−1^ using a spectrometer Jasco’s 4200 (Tokyo, Japan) with ATR attachment Pike Gladi ATR (Cottonwood, AZ, USA).

#### 2.2.5. UV-VIS Analysis

Changes of the physical properties as transmittance [%T] of poly(lactic acid) nonwoven fabrics, after coating modifications, were assessed using a double beam Jasco V-550 UV-VIS spectrophotometer (Tokyo, Japan) with integrating sphere attachment in the range: 200–800 nm.

#### 2.2.6. Filtration Parameters

Air permeability of poly(lactic acid) nonwoven fabrics were determined for one layer of the nonwoven sample and the test, and were performed according to EN ISO 9237:1998 standard, analogously as polypropylene nonwovens [65]. An FX 3300 TEXTEST AG (Klimatest, Poland) permeability tester was used. Air at a pressure of 100 Pascal and 200 Pascal was passed through a fabric area of 20 cm^2^ diameter for testing. An average of 10 values was taken to be the final value of the sample.

#### 2.2.7. Tensile Testing

Tensile testing of poly(lactic acid) nonwoven fabrics were carried out in accordance with the EN ISO 10319:2015-08 standard, analogously as polypropylene nonwovens [65]. A Tinius Olsen H50KS (Horsham, Pennsylvania, USA) tester was used. Stretching speed was 20 mm /min.

#### 2.2.8. Microbial Activity

The antibacterial activity of resulting poly(lactic acid)nonwoven fabrics were tested according to PN-EN ISO 20645:2006 (Textile fabrics—Determination of antibacterial activity—Agar diffusion plate test) against a colony of gram-negative bacteria: *Escherchia coli* (ATCC 25922) and gram-positive bacteria: *Staphylococcus aureus* (ATCC 6538), analogously as polypropylene nonwovens [65].

Antibacterial activity of modified poly(lactic acid) nonwoven fabrics were tested by the agar diffusion method using Muller Hinton medium agar. The test was initiated by pouring each agar onto sterilized Petri dishes and it was allowed to solidify. The surfaces of agar media were inoculated by overnight broth cultures of bacteria (ATCC 25922: 2.2 × 10^8^ CFU/mL, ATCC 6538: 1.9 × 10^8^ CFU/Ml). Samples of sterile PLA discs (10 mm) were charged with coating pastes with various amounts of fosfomycin (Table 7) and then discs with hybrids PLA/Fosfomycin were placed onto the inoculated agar and incubated at 37 °C for 24 h. The diameter of the clear zone around the sample was measured as an indication of inhibition of the microbial species. All tests were carried out in duplicate. Simultaneously, the same tests were carried out for control samples—samples of unmodified nonwoven fabrics.

## 3. Resultsand Discussion

The analysis of the biofunctionalized poly(lactide) nonwoven fabrics covered: scanning electron microscopy (SEM), attenuated total reflection Fourier transform infrared spectroscopy (ATR-FTIR), UV/VIS transmittance, and technical parameters: filtration parameters, tensile properties.The activity against representative gram-negative bacteria (*E. coli*) and gram-positive bacteria (*S. aureus*) of nonwoven fabrics modified by fosfomycin has been performed.

### 3.1. Scanning Electron Microscopy

Scanning electron microscopy (SEM) presents the routine technique for morphological investigations of polylactide nonwovens, both electrospin PLA [66,67,68,69,70,71] as well as melt-blown PLA [72,73,74,75,76,77,78,79,80,81] fibers. Scanning electron microscopy spectra of polylactide nonwoven and polylactide nonwoven coated with Fosfomycin modifier are presented in Figure 2 and Figure 3, respectively.

SEM spectra of PLA nonwovens present uniform randomly oriented nanofibers, with interconnected pores (space) between the nanofibers and relatively smooth surface. The average diameters of PLA nanofibers range from 3 to 8 µm, (Figure 2b,c).

Morphological changes of PLA after a surface deposition of fosfomycin lead to more random orientation of the fibers, with a more rough surface, covered with dots of the modifier (Figure 3c). These are much more subtle than described for the modification of polypropylene nonwoven with Ala-Ala^P^ [65].

### 3.2. ATR-FTIR Spectra

Comparison of ATR-FTIR spectra of fosfonomycin, polylactide nonwoven (PLA) and polylactide nonwoven charged with fosfonomycin (PLA-MOD) is presented in Figure 4, Figure 5 and Figure 6, respectively. Characteristic FTIR signals of fosfonomycin, and polylactide nonwoven (PLA)and polylactide–fosfonomycin nonwoven hybrid (PLA/fosfonomycin) are summarized in Table 4.

The IR spectrum ofpolylactide nonwoven coated by coating paste with 0.1%fosfomycinconcentrations (PLA-MOD) reveals the band derived from representative band of herbicidal agent, namely at 1410 cm^−1^.

### 3.3. UV/VIS Transmittance Spectra

Transmittance spectra [%T] of polypropylene nonwoven samples (PLA) and coated nonwoven sample with fosfomycin modifier (PLA-MOD) in the range λ = 200–800 nm are presented in Figure 7.

Transmittance (%T) spectra in the range λ = 200–800 nm of samples after modification revealed changes in macrostructure of the nonwoven after coating modification, expressed by the depress of transmittance ability in the all range of measuring spectra. Here, only the transmittance spectra of nonwoven coated by paste concentrations 0.1%fosfomycin (PLA-MOD) samples are shown, because the transmittance spectra’s nonwovens with different fosfomycin contents had the similar spectral characteristics and transmittance.

### 3.4. Technical Parameters

Technical parameters of coatedpolylactide nonwovens focused on filtration parameters and tensile strength. Filtration parameters expressed by the air permeability were detected for clean polylactide nonwoven and coatednonwovens. All these results are shown in Table 5 and indicated that modifications decreased, roughly two times (910 vs. 400–449 at 100 Pa and 880–890 at 200 Pa, respectively), the filtration properties of all nonwoven samples. Modified nonwoven samples with different fosfomycin contents had approximately a similar result for filtration properties. The fosfomycin content in the applied coating pastes on the polylactide nonwoven exhibit only slight effects on the filtration properties.

The testing results of tensile strength durability for stretching [kN/m] and relative elongation at maximum load [%] of polylactidenonwoven fabrics and modified polylactide (PLA) nonwovensare listed in Table 6.

The results show increased oftensile strength for modified PLA nonwovens (0.120 [kN/m]) in comparison with unmodified nonwoven (0.032 [kN/m]). There are no significant effects of modification on relative elongation of tested samples.

### 3.5. Antimicrobial Activity

The polylactide nonwoven (PLA), and polylactide nonwoven charged with fosfonomycin (PLA-MOD) were subjected to antimicrobial activity tests against gram-negative *E. coli* (ATCC11229) and gram-positive *S. aureus* (ATCC 6538) [84,85,86] (Table 7).

The results of these studies prove antimicrobial protection against different bacterial microorganisms of biofunctionalized materials. Even only a 0.005% concentration of Fosfomycin coating paste applied on polylactic acidnonwovens provides antimicrobial properties for *E. coli* and against *S. aureus* (Table 7), expressed by bold visible inhibition zones of bacterial growthin Petri dishes (Figure 8 and Figure 9).

## 4. Conclusions

As the consumption of disposable nonwoven products with short life increases, biodegradable polymers have great potential in use. On the other hand, demand for nonwovens is continually increasing. Polylactic acid (PLA) brings environmental benefits and can be used for the production of nonwoven fabrics.

This study focused on the functionalization of PLA nonwoven by the coating of bioactive compounds—Fosfomycin, into/on to the surface of their fibers. Fosfomycin is a low cost commercial antibiotic of natural occurrence, originally produced by certain types of *Streptomyces*, characterized by safety of use. The structure and mechanical properties of the obtained new biofunctionalized PLA nonwoven products were characterized by scanning electron microscopy (SEM), UV/VIS transmittance, FTIR spectrometry, and air permeability. This work has shown that our coating technology can be used to prepare new biodegradable or, “eco-friendly” biocomposite products with good mechanical properties. A significant attribute of the described process is uncomplicated, safe of implementation on an industrial scale and low production costs. All of the chemical agents and finishing agents which were used in this work are popular and easily available. The specific advantage of using hybrid materials based on polylactic acid (PLA) is the selected application in biomedical areas as an antibacterial material.

## Figures and Tables

**Figure 1 polymers-12-00768-f001:**
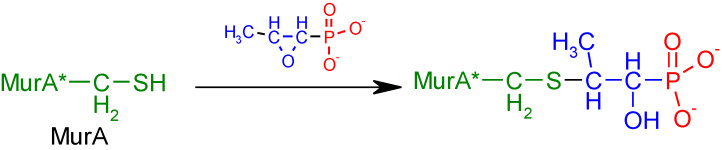
Mechanism of action of fosfomycin on the UDP-N-acetylglucosamine enolpyruvyl transferase enzyme (MurA).

**Figure 2 polymers-12-00768-f002:**
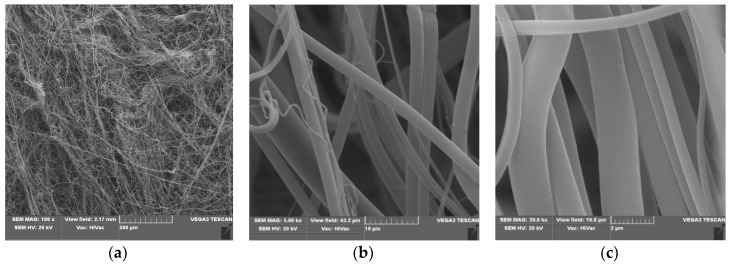
SEM result of Polylactide nonwoven (PLA), magnification 100× (**a**), 5000× (**b**) and 20,000× (**c**).

**Figure 3 polymers-12-00768-f003:**
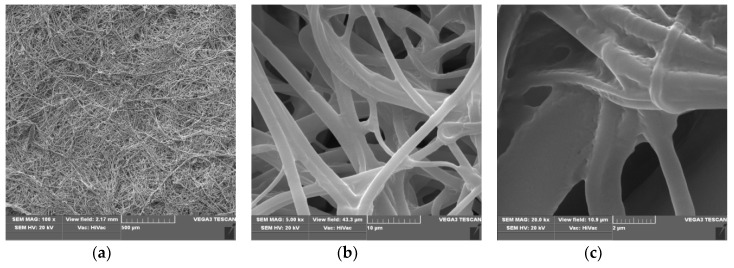
The SEM result of Polylactide nonwoven with fosfomycincoating (PLA-MOD), magnification 100× (**a**), 5000× (**b**) and 20,000× (**c**).

**Figure 4 polymers-12-00768-f004:**
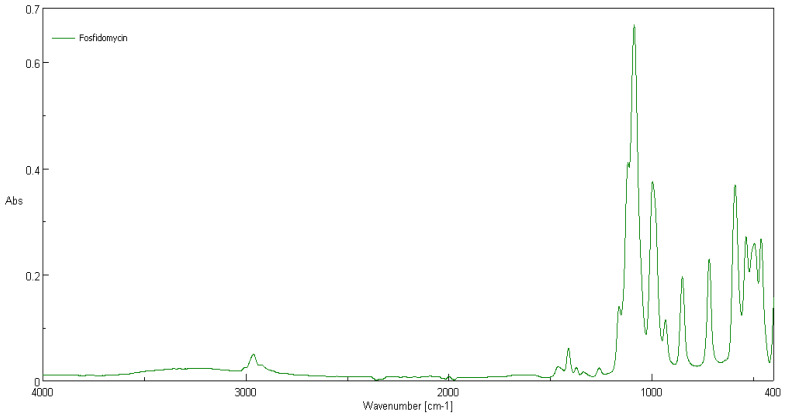
ATR-FTIR spectrum of fosfonomycin.

**Figure 5 polymers-12-00768-f005:**
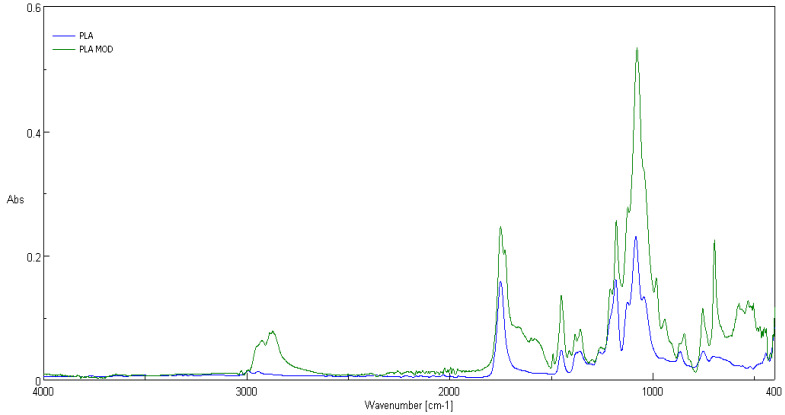
ATR-FTIR spectrum of the polylactide nonwoven (PLA) and nonwoven sample coated by paste concentrations 0.1%fosfomycin (PLA-MOD).

**Figure 6 polymers-12-00768-f006:**
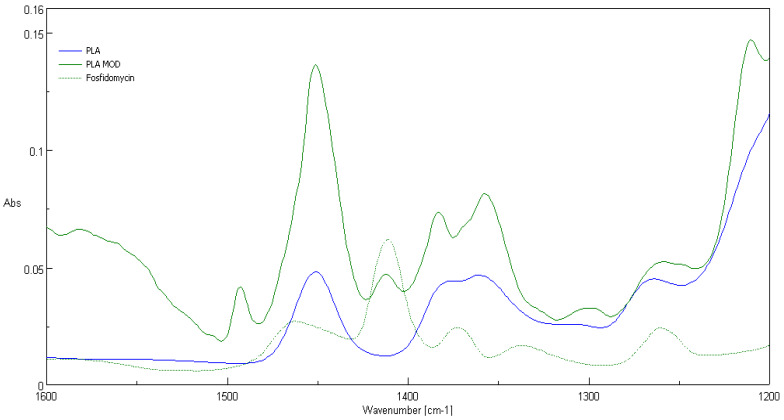
FTIR bands determined forfosfonomycin in wave number: 1410 cm^−1^ in spectra of the nonwoven sample coated by paste with Fosfomycin (PLA-MOD).

**Figure 7 polymers-12-00768-f007:**
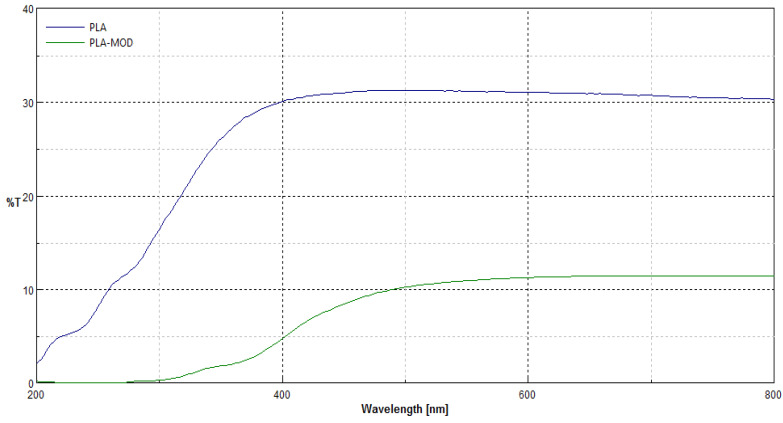
Comparison of transmittance spectra [%T] of polylactide nonwoven samples without (PLA) and with Fosfomycin coating (PLA-MOD) in the range λ = 200–800 nm.

**Figure 8 polymers-12-00768-f008:**
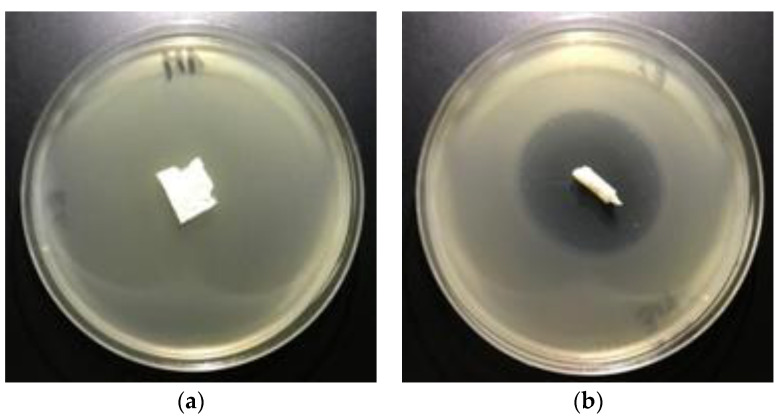
The nonwovens antimicrobial activity tests against *E. coli.* Inhibition zones of bacterial growth on Petri dishes, modified PLA nonwoven with fosfomycin coating pastes concentrations: **a**—0%; **b**—0.005%; **c**—0.01%; **d**—0.1%.

**Figure 9 polymers-12-00768-f009:**
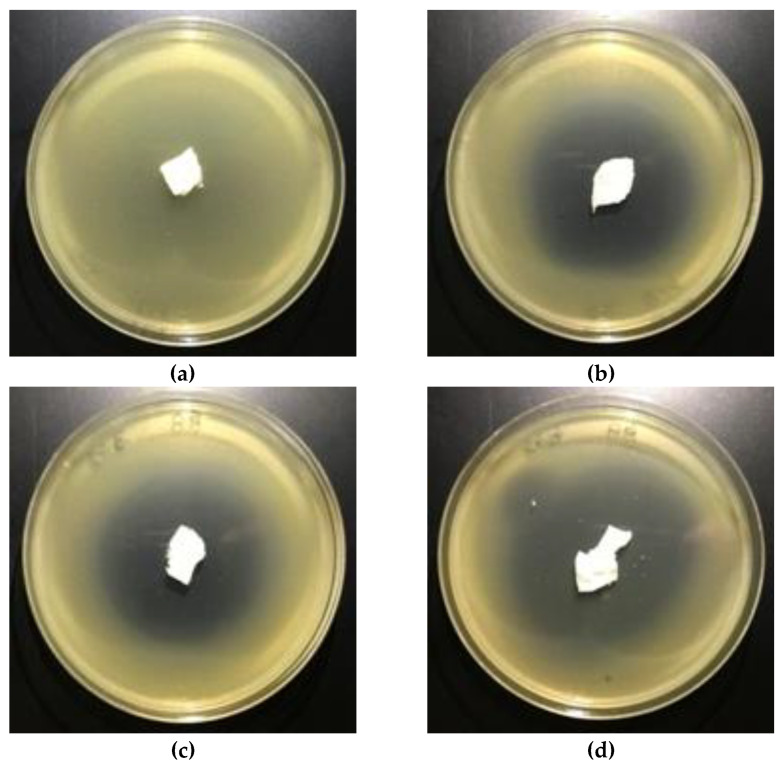
The nonwovens antimicrobial activity tests against *S. aureus.* Inhibition zones of bacterial growth on Petri dishes, modified PLA nonwoven with fosfomycin coating pastes concentrations: **a**—0%, **b**—0.005%; **c**—0.01%; **d**—0.1%.

**Table 1 polymers-12-00768-t001:** Structures, names, mode of actions and application area of representative biologically active functionalized phosphonic acid derivatives.

Name *(Abbreviation*) ^/a^	Structure ^/b^	Origin	Action/Application ^/c^	Ref.
**Phospho-glycine** (Gly^P^)	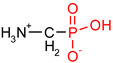	Primary PMG metabolite	Inhibition of prostate cancer cell growth (in vitro), phytotoxin	[30,31]
β-Ala^P^ (β-phosphono-alanine, 2-AEP, ciliatine)	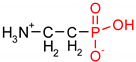	The first and most abundant natural AA^P^		[32,33]
**Phosphino-thricin** Glu^γP(Me)^ (phosphino-thricin, PPT)	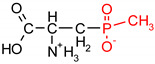	Produced by strains of *Streptomyces* herbicide	Inhibition of glutamine synthetase (E.C. 6.3.1.2)	[34]
**PMG** (Phosphono-Methyl-Glycine; Glyphosate)	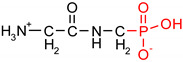	Artificial herbicide	inhibition of 5-enolpyruvyl-shikimic acid-3-phosphate synthase	[35,36]
**Alafosfalin** Ala-Ala^P^ (alaphosphin; alafosfalin)	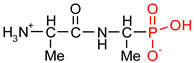	Artificial antibiotic, against gram-negative and gram-positive bacteria.	selective inhibition of alanine racemase (EC 5.1.1.1).	[37,38,39,40,41]
**Fosmidomycin** ((3-(Formyl-hydroxy-amino)-propyl)-phosphonic acid; The phosphonate antibiotic FR-31564	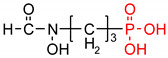	Produced by strains of *Streptomyces* A broad-spectrum antimicrobial agent	Inhibition of DXR (in vitro). A broad-spectrum antimicrobial agent currently applied for the malaria treatment.	[42,43,44,45,46]
**ϖ-Aminoalkyl-phosphonic acids** **(ϖ-AA^P^)**	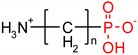	Artificial	Neuroactive acids	[47]
**Aminoalkyl-bisphosphonic acids** **(ϖ-AA^P,P^)**	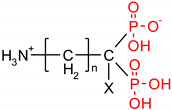 X = H, OH, halogen	Artificial	Inhibition of osteoclastic bone resorption	[48,49,50,51]
**Acyclic Nucleoside Phosphonates** (ANPs) Cidovir [HPMPC] (B = Cyt; X = OH; R = H); Adefovir [PMEA] (B = Ade; X = H; R = POM); Tenovir [PMPA] (B = Ade; X = H; R = POC)	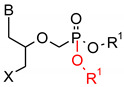	Artificial	Treatment of various DNA virus infections (*cidofovir*), hepatitis B (*adefovir*), and AIDS (HIV infections, *tenofovir*)	[52,53]
**Fosfomycin** (phosphomycin/phosphono-mycin)	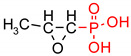	Fosfomycin - a broad-spectrum antibiotic produced by *Streptomyces* species.	Oral UTIs treatment. provide a useful option for the treatment of patients with pathogens with advanced resistance infections.	[54,55,56,57,58]

^a/^ Applied abbreviations according to the general rules elaborated by Kudzin at al. [59,60]. ^b/^ Applied in the form of free phosphonic acids or their salts. ^c/^ Applied enzymes/procedures abbreviations: DXR—deoxyxylulose phosphate reductoisomerase; UTIs—uncomplicated urinary tract infections; *5*-Enolpyruvylshikimate *3*-*phosphate* (EPSP) *synthase*. (EC 2.5.1.19).

**Table 2 polymers-12-00768-t002:** Technique processing parameters applied for preparation of Poly(lactic acid) nonwoven.

Processing Parameters	
Temperature of the extruder in zone 1	195 °C
Temperature of the extruder in zone 2	245 °C
Temperature of the extruder in zone 3	260 °C
Head temperature	260 °C
Air heater temperature	260 °C
Air flow rate	7–8 m³/h
Mass per unit area of nonwovens	102 g/m^2^
Polymer yields	6 g/min

**Table 3 polymers-12-00768-t003:** Component composition of used pastes [%].

Components	g	%
styrene-acrylic resin	6	6
thickening agent	1	1
wetting agent	3	3
water	90	90

**Table 4 polymers-12-00768-t004:** Characteristic FTIR bands determined for fosfomycin (Fosfm), polylactide (PLA) and the fosfomycin–polylactide nonwoven hybrid (PLA/Fosfm).

PLA [82]											
IR [ν/cm^−1^]; Intensity	2997; M	2947; M	1760; VS	1452; S	1348–1388; S	1368–1360; S	1270; S	1215–1185; VS	1130; S	1100–1090; VS, sh	1045; S
Assign- ment	ν_as_ CH_3_	ν_s_ CH_3_	ν C=O	δ_as_ CH_3_	δ_s_ CH_3_	δ_1_ CH + δ_s_ CH_3_	δCH+ ν COC	ν_as_ COC +r_as_CH_3_	r_as_CH_3_	ν_s_ COC	ν C-CH_3_
Fosfomycin (Fosfm) [83]											
FosfmNa_2_ (Nujol)											
IR [ν/cm^−1^]; Intensity		3010		1414 w			1270 w; 1260 w		1125 s; 1096 vs	1008 m	
Assign- ment		ν(C–H) (ring)		δ (CH_3_)			Ring breath		ν_a_ (PO_3_^2−^)	ν_a_ (PO_3_^2−^)	
FosfmCa × H_2_O (KBr)											
IR [ν/cm^−1^]; Intensity		3000 w		1423 w; 1419 sh			1262 vw		1095 vs	1017 m	
Assign- ment		ν(C–H) (ring)		δ (CH_3_)			Ring breath		ν_a_ (PO_3_^2−^)	ν_a_ (PO_3_^2−^)	

Legend: ν—stretching vibration, δ—deformation, sh = shoulder; s = symmetric; as = asymmetric; VS = very strong; S = strong; M = medium; w = weak.

**Table 5 polymers-12-00768-t005:** The air flow resistance of modified polylactide (PLA) nonwovens, according to: EN ISO 9237:1998.

Parameter		PLA	PLA-Fosfomycin [% Fosfomycin paste concentr.]		
			0.005%	0.01%	0.1%
**Average air permeability** **[mm/s], pressure decrease:**	100 Pa	910	442	440	449
	200 Pa	1677	880	876	891

The results have been measured in triplicate and presented as a mean value with ± deviation approximately 2%.

**Table 6 polymers-12-00768-t006:** Results of tensile strength test of modified polylactide (PLA) nonwovens.

Parameter	PLA	PLA-Fosfomycin [% fosfomycin paste concentr.]		
		0.005%	0.01%	0.1%
Tensile strength[kN/m]	0.032	0.117	0.120	0.115
Relative elongation at maximum load [%]	10.0	9.720	9.930	10.102

The results have been measured in triplicate and presented as mean value with ± deviation approximately 2%.

**Table 7 polymers-12-00768-t007:** Results of tests on the antibacterial activity of fosfomycin modified nonwovens.

Sample No.	*Fosfomycin* *on PLA* *nonwovens*	Bacterial Average Inhibition ZoneGrowth for Bacteria (mm)	
	Fosfomycin Coating Pastes Concentrations (%)	*E. coli*	*S. aureus*
1	0	0	0
2	0.005	3–4	5
3	0.01	4–5	6
4	0.1	5–6	6

Concentration of inoculum (bacterial suspension) amount of live bacteria. - *E. coli*: *CFU/mL =* 2.2 × 10^8^. - *S. aureus*: *CFU/mL* = 1.9 × 10^8^.

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
