# Peer review of "Biofunctionalization of Textile Materials. 2. Antimicrobial Modification of Poly(lactide) (PLA) Nonwoven Fabricsby Fosfomycin"

_polymers, 2020, doi:10.3390/polym12040768_

Round 1

Reviewer 1 Report

This paper is interesting with moderate novelty in the field. However, authors should improve the paper according to following lines:

1- XRD of samples should be provided before and after treatment.

2- Thermal properties and flammability of samples should be tested. Phosphonate based derivative materials have been recently reported for cotton as flame proof and thermal resistant component. These studies was not reported in this article. Also the effect of coating on these properties of PLA should be reported and compared with those studies.

3- The effect of coating on contact angle and moisture absorption properties of PLA should also be tested. 

Reviewer 2 Report

Manuscript need to be strengthen by improving the discussion part. 

Results must be validated and discussed with suitable references.

SEM images must be explained clearly with the other results obtained.

Round 2

Reviewer 1 Report

The paper is acceptable now.

Reviewer 2 Report

Author had corrected as per the given suggestion. 

Results and discussion part has been validated.

Manuscript can be considered for publication.